# MRI Characteristics Accurately Predict Biochemical Recurrence after Radical Prostatectomy

**DOI:** 10.3390/jcm9123841

**Published:** 2020-11-26

**Authors:** Cécile Manceau, Jean-Baptiste Beauval, Marine Lesourd, Christophe Almeras, Richard Aziza, Jean-Romain Gautier, Guillaume Loison, Ambroise Salin, Christophe Tollon, Michel Soulié, Bernard Malavaud, Mathieu Roumiguié, Guillaume Ploussard

**Affiliations:** 1Department of Urology, CHU Toulouse-IUCT Oncopole, 31400 Toulouse, France; cecile.manceau2@gmail.com (C.M.); marine_lsrde@hotmail.fr (M.L.); soulie.m@chu-toulouse.fr (M.S.); bernard.malavaud@me.com (B.M.); roumiguie_mathieu@yahoo.fr (M.R.); 2Department of Urology, La Croix du Sud Hospital, 31130 Quint Fonsegrives, France; jbbeauval@gmail.com (J.-B.B.); c.almeras@gmail.com (C.A.); gautierjr@hotmail.fr (J.-R.G.); guillaumeloison@gmail.com (G.L.); ambroise.salin@gmail.com (A.S.); tol@club-internet.fr (C.T.); 3Department of Radiology, Institut Universitaire du Cancer Toulouse Oncopole, 31400 Toulouse, France; Aziza.Richard@iuct-oncopole.fr

**Keywords:** prostate cancer, radical prostatectomy, targeted biopsies, magnetic resonance imaging (MRI), systematic biopsies, biochemical recurrence, prostate cancer prognosis, imaging-based risk classification

## Abstract

Background: After radical prostatectomy (RP), biochemical recurrence (BCR) is associated with an increased risk of developing distant metastasis and prostate cancer specific and overall mortality. Methods: The two-centre study included 521 consecutive patients undergoing RP for positive pre-biopsy magnetic resonance imaging (MRI) and pathologically proven prostate cancer (PCa), after which a combination scheme of fusion-targeted biopsy (TB) and systematic biopsy was performed. We assessed correlations between MRI characteristics, International Society of Urological Pathology (ISUP) grade group in TB, and outcomes after RP. We developed an imaging-based risk classification for improving BCR prediction. Results: Higher Prostate Imaging and Reporting and Data System (PI-RADS) score (*p* = 0.013), higher ISUP grade group in TB, and extracapsular extension (ECE) on the MRI were significantly associated with more advanced disease (pTstage), higher ISUP grade group (*p* = 0.001), regional lymph nodes metastasis in RP specimens (*p* < 0.001), and an increased risk of recurrence after surgery. A positive margin status was significantly associated with ECE-MRI (*p* < 0.001). Our imaging-based classification included ECE on MRI, ISUP grade group on TB, and PI-RADS accurately predicted BCR (AUC = 0.714, *p* < 0.001). This classification had more improved area under the curve (AUC) than the standard d’Amico classification in our population. Validation was performed in a two-centre cohort. Conclusions: In this cohort, PI-RADS score, MRI stage, and ISUP grade group in MRI-TB were significantly predictive for disease features and recurrence after RP. Imaging-based risk classification integrating these three factors competed with d’Amico classification for predicting BCR.

## 1. Introduction

Prostate cancer (PCa) is the second most common cancer worldwide and the fifth leading cause of death from cancer among men [1].

Radical prostatectomy (RP) is a treatment for patients with localized disease and with at least a ten years of life expectancy [2]. The goal is the eradication of the tumor by removing the entire prostate with an undetectable serum prostatic specific antigen (PSA). Some patients have measurable PSA during routine post-surgery follow-ups, which characterize biochemical recurrence (BCR). BCR is associated with an increased risk of developing distant metastasis, PCa-specific mortality and, to a lesser extent, overall mortality [3]. Recent multivariate analysis [3] suggested a new classification for patients experiencing BCR that differentiates patients with low or high risk of clinical progression based on PSA-doubling time, interval to biochemical failure, and prostatectomy gleason score. Predicting BCR could guide optimal treatment decisions and surgery. To date, prediction of BCR is still based on risk classification including PSA, grade group, and clinical stage without incorporating magnetic resonance imaging (MRI) criteria [2].

However, during the last decade, MRI has emerged as a powerful imaging tool for diagnosis, staging, and preoperative planning. Since 2018, updated prostate cancer guidelines [2] recommend the realization of a multi-parametric MRI prior to biopsies to localise suspicious areas that could be targeted. In the case of positive MRI (PI-RADS 3 or more), targeted biopsies (TB) should be directed to all visible lesions. This technique highlights TB as superior over systematic biopsies (SB) for the detection of clinically significant prostate cancer [4,5,6,7] (ISUP > 1 or 2 depending on the study and cancer core length > 6 mm).

Recent studies have explored and confirmed the utility of prostate MRI to improve detection of significant PCa and to make a risk stratification for locally advanced disease. Nevertheless, other studies explored prostate mpMRI for prognosis by predicting BCR, but presented contradicting results [8,9,10,11].

To our knowledge, no study has assessed the impact of a targeted biopsy on BCR prediction concomitantly with the other MRI criteria, alone or in combination.

The aim of this present study was to evaluate the performance of MRI criteria and an MRI-guided biopsy pathway for predicting BCR after RP for PCa.

## 2. Experimental Section

### 2.1. Patient Selection, Assessment, Treatment, and Follow-Up

The two-centre study population consisted of 521 consecutive patients undergoing RP for positive pre-biopsy MRI and pathologically-proven PCa, after which a combination scheme of fusion targeted biopsy (TB) and systematic biopsy (SB) was performed, between 2015 and 2019. Patients who had adjuvant treatment without BCR were excluded.

MRI lesions were submitted to targeted biopsy using real-time transrectal ultrasound (TRUS) guidance via a software registration system with elastic fusion (Koelis^®^ system, Koelis Inc., Princeton, NJ, United States). The number of targeted and systematic cores taken for each suspicious lesion on mpMRI was chosen at the physician’s discretion. At least two TBs per suspicious lesion were taken. TB and SB were performed at the same time during the biopsy procedure, and the operator was aware of clinical-biological and mpMRI results. All operators were experienced in fusion biopsy procedures (the same device in both centres, and personal experience before study entry > 60 TB procedures). In biopsies, a grade group was performed for each area (*n* = 6) in case of SB according to recommendations [12], and the total grade group was the worst; in case of targeted biopsies and radical prostatectomy, a grade group was assessed for each focus, and the final grade group was the one of the index lesion, the lesion with the highest grade group. Indication for RP was taken according to EAU guidelines [2]. Unless stated otherwise, RPs were performed by high volume surgeons. Biopsy and RP specimens were evaluated by senior dedicated uropathologists. Data from clinical evaluations, biopsies, RP specimens, and follow-ups were recorded in a prospective database.

### 2.2. MRI Protocol

The imaging protocol consisted of multi-planar T2-weighted images, diffusion-weighted imaging, dynamic contrast-enhanced MRI, and T1-weighted images with fat suppression according to the European Society of Urogenital Radiology guidelines [13]. Both institutions used a 1.5-T MR unit and a 16-channel coil. No endorectal coil was used. The maximal b-value used for diffusion-weighted imaging was b 2000. The mpMRI images were scored and reported according to Prostate Imaging-Reporting and Data System v.2 (PI-RADS) [14] using the five-point scale. Extracapsular extension (ECE) was suspected due to evidence of capsular overshoot, bulging, or contact extension. Five expert uroradiologists read the MRIs, and all had more than two years of experience before study entry. MRI data prior to 2016 were re-reviewed according to this updated PI-RADS version.

### 2.3. BCR

After RP, PSA was expected to be undetectable within six weeks. Biochemical follow-up was standardized with a PSA test at six weeks, three months, six months, and then every six months after surgery for five years. According to the guidelines of the French and American Urological Association Localized Prostate Cancer Update Panel report [15,16], BCR was defined as a serum PSA ≥ 0.2 ng/mL with a confirmatory value of ≥ 0.2 ng/mL, or a single PSA ≥ 0.4 ng/mL, or by the receipt of salvage therapy, specifically due to an increasing postoperative PSA.

### 2.4. Statistical Analyses

We collected the clinical data (age and digital rectal examination), biological data (pre-operative PSA, post-operative PSA, and follow up), MRI information (PI-RADS V2 category, prostate volume, ECE on MRI, MRI lesion number, and MRI lesions size), and pathological findings, such as ISUP grade group in TB and SB, along with pTN stage in the overall population.

The primary endpoint was the time to BCR. We assessed correlations between MRI characteristics (PI-RADS, lesion diameter and number, and MRI stage), ISUP grade group in TB, and outcomes after RP. The qualitative data were tested using a chi-square test or Fisher’s exact test as appropriate, and the continuous data were tested using Student’s *t*-test. The Mann–Whitney test was used in case of abnormal distribution. We used the Kaplan–Meier method to study BCR free survival and survival curves among the groups, compared using the log-rank test. Univariate regression models were performed to evaluate the association between variables and biochemical recurrence. The limit of statistical significance was defined as *p* < 0.05. SPSS 22.0 (IBM Corp. Released 2013, IBM SPSS Statistics for Mac Version 22.0, Armonk, NY, United States.) software was used for analysis.

## 3. Results

### 3.1. Population Characteristics

Patient characteristics are shown in Table 1. The mean patient age was 64.9 years. The mean PSA and PSA density were 10.26 ng/mL (median = 8) and 0.24 (median = 0.18) ng/mL/gr, respectively. Clinical T2-T3 was reported in 34.3% of cases. According to the classification proposed by D’Amico et al. [17], 20.0% were in the high-risk group based on SB and TB results. In the overall cohort, a high grade was reported in 9.0% of RP specimens. Overall, 13.1% of patients had regional lymph node metastasis, and 22.1% had positive margins. During a median follow-up time of 12.4 months [1 to 53 months], 9.4% of patients experienced BCR.

### 3.2. BCR-Free Survival According to PI-RADS Score

During the follow-up period, the rate of BCR at 12 months was 0.014, 0.056, and 0.107 among patients with PI-RADS of 3, 4 and 5, respectively. The two-year BCR free survival curves (Figure 1) were significantly different according to the PI-RADS score (*p* = 0.006). A higher PI-RADS score was associated with a higher tumor stage in RP (*p* = 0.013), regional lymph node metastasis (*p* < 0.001), and higher ISUP grade group on RP (*p* = 0.001), yet showed no positive margin (*p* = 0.053).

### 3.3. BCR-Free Survival According to ECE on MRI

Overall, 15.9% of patients had pre-operative ECE on MRI. For these patients, 79.8% had pT3-4 stage in RP specimens (*p* < 0.001), 30.1% had regional lymph node metastasis (*p* < 0.001), and 36.1% had a positive margin (*p* < 0.001) (Table 2). BCR free survival curves were significantly different according to the MRI stage (no ECE versus ECE on MRI, *p* < 0.001) (Figure 1).

### 3.4. BCR-Free Survival According to Number of MRI Lesions

The rate of BCR at 12 months was 0.067 and 0.081 among patients having one or more than two lesions on the MRI, respectively (*p* = 0.684) (Table 2).

The number of MRI lesions was not statistically correlated with BCR-free survival (*p* = 0.912) (Figure 1). In RP specimens, the number of lesions was not correlated with regional lymph node metastasis, positive margin, or ISUP grade group in RP. Nonetheless, it was positively correlated with a higher pT stage (*p* < 0.001).

### 3.5. BCR-Free Survival According to Maximal Lesion Diameter

We found that the MRI lesion diameter significantly predicted BCR-free survival using the Kaplan–Meier method (*p* = 0.009) (Figure 1). Moreover, the pT stage was positively correlated with the MRI lesion diameter (*p* = 0.003). No significant association was reported regarding regional lymph node metastasis or positive margin (*p* = 0.269 and *p* = 0.262, respectively).

### 3.6. BCR-Free Survival According to ISUP Grade Group in Targeted Biopsy

Overall, a higher ISUP grade group on TB was significantly associated with a higher pT stage in RP specimens (*p* < 0.001), regional lymph node metastasis (*p* < 0.001), higher ISUP grade group in RP (*p* < 0.001), and positive margin status (*p* = 0.001, Table 2). On survival analysis, the ISUP grade group was also significantly associated with an increased risk of recurrence (*p* = 0.001, Figure 1).

### 3.7. MRI Imaging-Based Risk Classification

Finally, we developed imaging-based classification in the centre 1 cohort (*n* = 299), incorporating ISUP grade group in TB, PI-RADS, and ECE on MRI as predictors for BCR. Different cutoffs have been chosen based on their predictive value in univariable analysis using the HR of each value. We have chosen not to incorporate the diameter of the lesions in our model due to the low availability of this information in current practice. In our cohort, this information was not available for almost 20% of the patients. This classification was then validated in the centre 2 cohort (*n* = 222).

This imaging-based risk classification included three risk groups as follows:Low risk, which includes no ECE on MRI, ISUP grade group 1–2 in TB, and PI-RADS < 5Intermediate risk, which includes PI-RADS = 5 or ISUP grade group 3 in TB with no ECE MRIHigh-risk, which includes ECE on MRI or ISUP grade group 4–5 in TB (regardless of the PI-RADS)

This classification was significantly correlated with the risk of BCR in both centres, with *p* = 0.021 in centre 1 and *p* < 0.001 in centre 2 (Appendix A).

In the overall population, this classification was significantly correlated with the risk of BCR (*p* < 0.001) (Figure 2). Intermediate and high-risk patients demonstrated a 70% increased risk of recurrence (HR = 1.72) and by 7.2-fold (HR = 7.17), respectively, when compared with low risk patients. The one-, two-, and three-year RFS rates were 97.4%, 93.4%, and 85.4% in the low risk group (*n* = 17.3% of the overall cohort); 94.2%, 90.4%, and 77.1% in the intermediate risk group (*n* = 40.5%); and 85.0%, 77.0%, and 61.4% in the high-risk group (*n* = 42.2%), respectively.

In the centre 1 cohort, the AUC for predicting BCR was 0.714 for the imaging-based classification compared with 0.710 for the d’Amico classification. In the centre 2 cohort, the AUC for predicting BCR was 0.676 for the imaging-based classification compared with 0.655 for the d’Amico classification.

## 4. Discussion

Recently, the PCa diagnostic pathway has drastically evolved; mpMRI changed clinically localized prostate cancer diagnosis [6], with a continuous improvement of the quantity and quality of information. The imaging-based strategy has been proven to improve the detection of clinically significant prostate cancer in various studies [4,5,6,7] with the added value of TB [18,19,20]. The added value of MRI has been extensively assessed for diagnostic purposes. Further studies are yet to be undertaken regarding prognosis assessment and post-therapeutic outcome predictions. Few studies have correlated MRI findings with recurrence risk after RP. Moreover, preoperative risk classification might differ between patients diagnosed with systematic biopsies alone without mpMRI and those with mpMRI and targeted biopsy, given the supplementary data obtained by targeting MRI lesions. Actual risk classifications based only on digital rectal examination, PSA, and systematic biopsies could be improved by this imaging-guided information. Novel risk models incorporating clinical parameters and MRI data have been suggested to perform significantly better than risk calculators and classification validated in the pre-MRI era [21].

The main early surrogate of cancer cure after surgery is the absence of biological recurrence. BCR is linked to more advanced disease, and has been associated with increased rates of metastasis and prostate cancer specific mortality [22]. Given the recent evolution of the biopsy diagnosis pathway, we are yet to study cohorts of patients undergoing MRI-TB with a sufficiently long post-surgical period, enabling the assessment of clinically strong endpoints, such as metastasis-free survival or overall survival. Thus, BCR should be considered to date as an interesting surrogate. A preoperative model for the prediction of BCR after RP, including MRI data and TB, could be of great value for improving risk stratification and patient counseling before treatment decision-making.

Different MRI parameters have been correlated to BCR after surgery, such as index tumor volume [23], Likert score [24], and ADC [25]. Preoperative predictive models for disease recurrence have been previously proposed including MRI data [8,10,25,26], yet, to our knowledge, no study has incorporated the finding on targeted biopsy.

Park SY et al. suggested that PI-RADS V2 classification was an independent factor of BCR after RP [9]. In a recent study, Faiena et al. showed in a cohort of ISUP grade group 2 lesions that a PI-RADS 5 lesion predicted adverse features and biochemical recurrence-free survival [27], in line with our findings.

Furthermore, Ho et al. showed in a cohort of 370 patients undergoing RP that MRI suspicion score and the suspicion of extra prostatic extension on MRI were both predictive of BCR after surgery. The addition of these factors to standard clinical factors significantly improved BCR prediction. The main limitation of these studies was that findings of TB were not included into predictive models. Only standard biopsy results were assessed, and therefore, the final grade prediction might be partially inaccurate.

In our study, we chose to only include patients who had a pre-biopsy positive MRI and were undergoing fusion TB. Thus, we are able to assess the predictive value of a complete imaging-guided pathway, including MRI characteristics and TB results.

We showed that the PI-RADS score, ECE on MRI, and ISUP grade group on TB were predictive factors for BCR after surgery. We aimed to build a risk classification only based on imaging-based features. Consequently, the proposed three-group classification has been built only on predictive factors of BCR: the PI-RADS score, ECE on MRI, and ISUP grade group on TB. We chose to exclude standard features, such as PSA, clinical stage, and ISUP grade group on standard biopsy, in order to show that imaging data combined with MRI-TB could be clinically relevant to predict post-RP outcomes. This imaging-based classification outperformed the standard d’Amico classification for BCR prediction in a selected cohort of patients undergoing MRI-TB.

We have validated the results externally in a second centre with different urologists, radiologists, and uropathologists, but using the same fusion biopsy device, with a comparable added value of the imaging-based classification as compared to the d’Amico classification. The next step will be to repeat our investigation in a larger cohort of patients, with a longer follow-up, more events, and the combination of all clinical, biological, radiological, and pathological features into nomograms for BCR prediction purposes.

Several limitations have to be emphasized. Our study cohort only included patients treated by surgery for localized prostate cancer. Thus, patients undergoing radiotherapy, brachytherapy, or active surveillance were not included, which led to a selection bias. In addition, this model is only useful if MRI and fusion biopsy are available, which, according to the recommendations, are part of the standard of care.

Our median follow-up was relatively short, limited to 12.4 months with 9.4% BCR. This relatively short follow-up could explain the low number of recurrences observed. Early BCR is known to have a high risk of metastasis and PCa-specific mortality, together associated with oncological outcomes [3,28]. On one hand, a more extensive follow-up would have been more appropriate. On the other hand, the prediction of early BCR remains accurate, as about two-thirds of PSA recurrences occur within two years of surgery. Conducting further studies will be necessary to validate these preliminary results using multicenter collaborations and a longer follow-up series.

No central review was obtained for MRI images and pathology specimens. Radiologists over-staged lesions for 11 patients, incorrectly claiming an extracapsular extension. Low sensibility of extracapsular extension on MRI is known [29], but over-staging was not studied and could present as a limitation in our model. However, all radiologists and biopsy operators involved in our study were highly experienced in computer-based fusion devices and beyond their learning curves since the beginning of the study period. The same fusion computer-assisted software was used in the two institutions, which reduced interpretation biases. Moreover, this elastic registration system has been proven to improve precision for targeting and correlated with improved detection of clinically significant PCa compared with the cognitive fusion method [30,31].

## 5. Conclusions

MRI has been demonstrated to be a valuable tool for improving diagnosis of clinically significant PCa. It could also optimize recurrence risk prediction before treatment decision-making. We found in the present series that the PI-RADS score, T stage on MRI, and ISUP grade group in TB accurately predicted the risk of recurrence after surgery. When incorporating these factors into an imaging-based risk classification, it outperformed the standard d’Amico classification in this cohort of patients undergoing fusion TB after a positive pre-biopsy MRI. This classification has been validated in a separate cohort, nevertheless, external validation in series with longer follow-ups are needed.

## Figures and Tables

**Figure 1 jcm-09-03841-f001:**
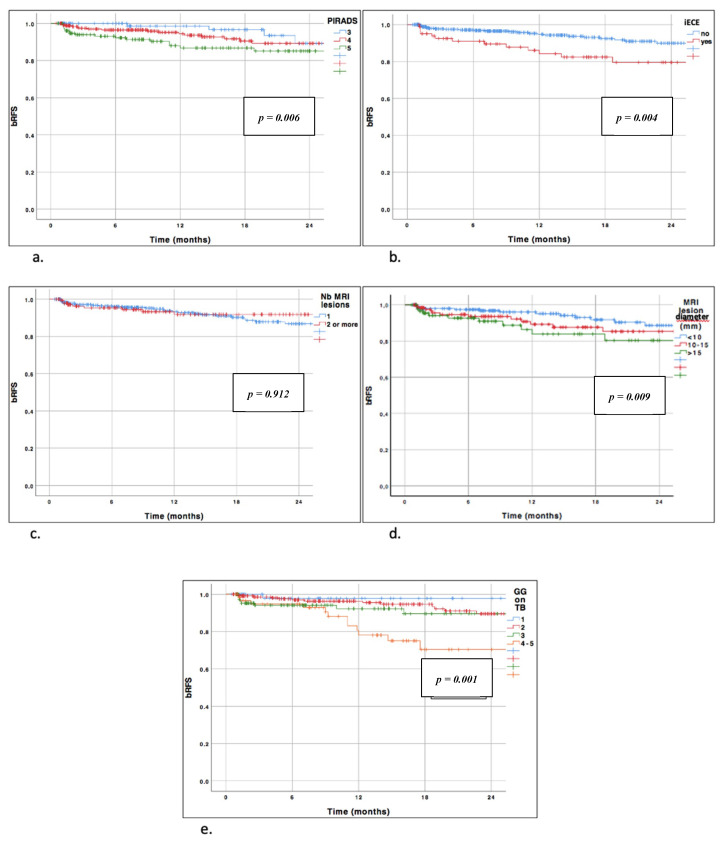
Biochemical recurrence free curves stratified by MRI characteristics. (**a**) Biochemical recurrence free curves stratified by PI-RADS. Log rank test *p* = 0.006. (**b**) Biochemical recurrence free curves stratified by IRM extracapsular extension. Log-rank test *p* = 0.004. (**c**) Biochemical recurrence free curves stratified by number of MRI lesions. Log-rank test *p* = 0.912. (**d**) Biochemical recurrence free curves stratified by MRI lesion diameter. Log-rank test *p* = 0.009. (**e**) Biochemical recurrence free curves stratified by grade group on targeted biopsy. Log-rank test *p* = 0.001. bRFS = biochemical recurrence free survival; iECE = extracapsular extension on MRI; MRI: magnetic resonance imaging; Nb MRI lesions = number of MRI lesions; GG on TB = grade group on targeted biopsies; PI-RADS: prostate imaging and reporting and data system.

**Figure 2 jcm-09-03841-f002:**
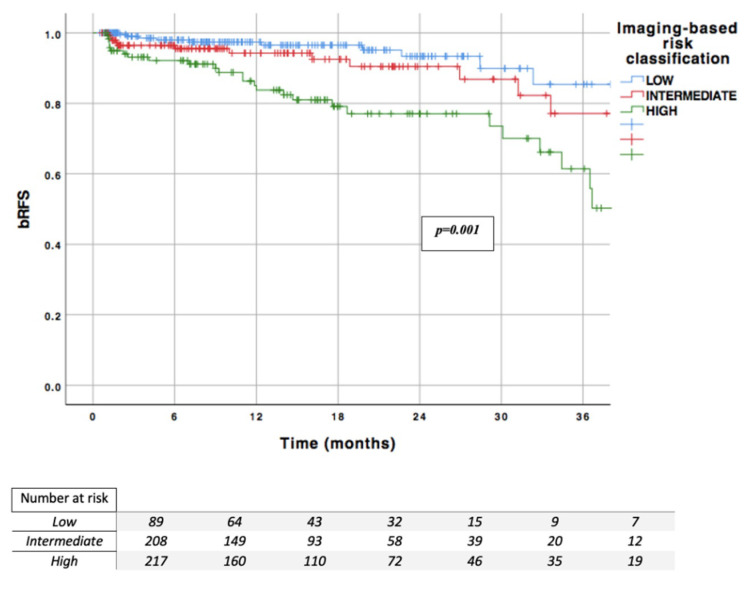
Imaging-based risk classification; bRFS: biochemical recurrence free survival.

**Table 1 jcm-09-03841-t001:** Patient characteristics.

	Overall
	*n* = 521
Age (years):	
Mean	64.86
Median (Range)	65.70 (46.80–77.90)
PSA (ng/mL):	
Mean	10.26
Median (Range)	8 (0.65–69)
PSAD:	
Mean	0.24
Median (Range)	0.18 (0.03–1.28)
Clinical stage:	
T1	342 (65.6%)
T2	168 (32.2%)
T3	11 (2.1%)
PI-RADS	
3	103 (19.8%)
4	260 (49.9%)
5	158 (30.3%)
No of MRI lesions	
1	373 (71.6%)
2	121 (23.2%)
≥3	27 (5.2%)
MRI Maximal lesion diameter (mm):	
<10	211 (40.5%)
10–15	130 (25%)
>15	91 (17.5%)
Missing	89 (17.0%)
MRI stage:	
No ECE on MRI	436 (83.7%)
ECE on MRI	83 (15.9%)
Missing	2 (0.4%)
ISUP grade group TB	
Negative biopsies	57 (11.0%)
1	60 (11.5%)
2	235 (45.1%)
3	108 (20.7%)
4–5	61 (11.7%)
ISUP grade group TB	
Negative biopsies	82 (15.7%)
1	134 (25.7%)
2	203 (39.0%)
3	64 (12.3%)
4–5	38 (7.3%)
Pathological stage pT:	
pT2	245 (47.0%)
pT3a	185 (35.5%)
pT3b	91 (17.5%)
Regional lymph nodes pN	
pNx	68 (13.1%)
pN0	403 (77.4%)
pN1	50 (9.6%)
ISUP grade group RP:	
1	16 (3.1%)
2	269 (51.6%)
3	189 (36.3%)
4–5	47 (9.0%)
Margin:	
Negative	406 (77.9%)
Positive	115 (22.1%)
Biochemical recurrence:	
YES	49 (9.4%)
NO	465 (89.3%)
Missing	7 (1.3%)

ECE: extracapsular extension; ISUP: International Society of Urological Pathology; MRI: magnetic resonance imaging; PSA: prostate specific antigen; PSAD: prostate specific antigen density; PI-RADS: prostate imaging and reporting and data system; pN: pathological lymph node invasion; pT: pathological tumoral stage; TB: targeted biopsy.

**Table 2 jcm-09-03841-t002:** Association between MRI characteristics and Pathological features in radical prostatectomy specimens and Biochemical recurrence.

	Overall	pT Stage in RP				pN1		ISUP Grade Group in RP					Margin R1	
		pT2 245 (47.0%)	pT3a 185 (35.5%)	pT3b–pT4 91 (17.5%)		50 (9.6%)		1 16 (3.1%)	2 269 (51.6%)	3 189 (36.3%)	4–5 47 (9.0%)		115 (22.1%)	
PIRADS					*p* = 0.013		*p* < 0.001					*p* = 0.001		*p* = 0.053
3	(*n* = 103)	53 (51.5%)	35 (34.0%)	15 (14.6%)		2 (1.9%)		5 (4.9%)	57 (55.3%)	34 (33.0%)	7 (6.8%)		28 (27.2%)	
4	(*n* = 260)	135 (51.9%)	87 (33.5%)	38 (14.6%)		22 (8.5%)		6 (2.3%)	155 (59.6%)	78 (30.0%)	21 (8.0%)		46 (17.7%)	
5	(*n* = 158)	57 (36.1%)	63 (39.9%)	38 (24.1%)		26 (16.5%)		5 (3.2%)	57 (36.1%)	77 (48.7%)	19 (12.0%)		41 (25.9%)	
ECE on MRI					*p* < 0.001		*p* < 0.001					*p* <0.001		*p* < 0.001
No	(*n* = 436)	232 (53.2%)	152 (34.9%)	52 (11.9%)		25 (5.7%)		16 (3.7%)	243 (55.7%)	149 (34.2%)	28 (6.4%)		85 (19.5%)	
Yes	(*n* = 83)	11 (13.2%)	33 (39.8%)	39 (47.0%)		25 (30.1%)		0 (0.0%)	25 (30.1%)	39 (47.0%)	19 (22.9%)		30 (36.1%)	
No of MRI lesions					*p* < 0.001		*p* = 0.269					*p* = 0.216		*p* = 0.262
1	(*n* = 373)	174 (46.6%)	132 (35.4%)	67 (18.0%)		34 (9.1%)		11 (2.9%)	191 (51.2%)	138 (37.0%)	33 (8.9%)		87 (23.3%)	
2	(*n* = 121)	60 (49.6%)	41 (33.9%)	20 (16.5%)		14 (11.6%)		4 (3.3%)	59 (48.8%)	44 (36.4%)	14 (11.6%)		25 (20.7%)	
≥3	(*n* = 27)	11 (47.0%)	12 (44.4%)	4 (14.8%)		2 (7.4%)		1 (3.7%)	19 (70.4%)	7 (25.9%)	0 (0.0%)		3 (11.1%)	
MRI Maximal lesion diameter (mm):					*p* = 0.003		*p* = 0.070					*p* = 0.045		*p* = 0.125
<10	(*n* = 211)	109 (51.7%)	72 (34.1%)	30 (14.2%)		14 (6.6%)		5 (2.4%)	120 (56.9%)	65 (30.8%)	21 (10.0%)		46 (21.8%)	
10–15	(*n* = 130)	60 (46.2%)	44 (33.8%)	26 (20.0%)		16 (12.3%)		4 (3.1%)	59 (45.4%)	56 (43.1%)	11 (8.5%)		25 (19.2%)	
>15	(*n* = 91)	28 (45.6%)	35 (38.5%)	28 (30.8%)		16 (17.6%)		2 (2.2%)	35 (38.5%)	40 (44.0%)	14 (15.4%)		28 (30.8%)	
ISUP grade group in TB					*p* < 0.001		*p* < 0.001					*p* < 0.001		*p* = 0.001
1	(*n* = 60)	33 (55.0%)	19 (31.7%)	8 (13.3%)		1 (1.7%)		3 (33%)	44 (73.3%)	2 (3.3%)	3 (5.0%)		12 (20%)	
2	(*n* = 235)	123 (35.2%)	82 (34.9%)	30 (12.8%)		12 (5.1%)		4 (1.7%)	152 (64.7%)	72 (30.6%)	7 (3.0%)		42 (17.9%)	
3	(*n* = 108)	38 (35.2%)	41 (38.0%)	29 (26.9%)		19 (17.6%)		2 (1.9%)	29 (26.9%)	69 (63.9%)	8 (7.5%)		27 (25%)	
4–5	(*n* = 61)	14 (23.0%)	24 (39.3%)	23 (37.7%)		15 (24.6%)		0 (0.0%)	6 (9.8%)	29 (47.5%)	26 (42.6%)		26 (42.6%)	

ECE: extracapsular extension; ISUP: International Society of Urological Pathology; MRI: Magnetic resonance imaging; PSA: Prostate specific antigen, PSAD: Prostate specific antigen density; PI-RADS: Prostate Imaging and Reporting and Data System; RP: radical prostatectomy; TB: targeted biops.

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
