# Peer review of "MRI Characteristics Accurately Predict Biochemical Recurrence after Radical Prostatectomy"

_jcm, 2020, doi:10.3390/jcm9123841_

Round 1
Reviewer 1 Report
In this paper the authors investigated an MRI-based risk classification (including GG on TB, PIRADS and ECE on MRI) as predictor for BCR.
The idea is not novel since several nomograms has been developed lately but it is the first using target biopsy. Several issues deserves to be pointed out:
- Several % in table 1 are not consistent (e.g. MRI stage and BCR)
- Table 2 is hard to read in this fashion, I suggest to shift it in a vertical way in order to use a bigger font.
- “the AUC for predicting BCR was 0.714 for the imaging-based classification compared with 0.710 for the d’Amico classification. In centre 2 cohort, the AUC for predicting BCR was 0.676 for the imaging-based classification compared with 0.655 for the d’Amico classification.” àoverall, the suggested classification did not impact much on the AUC in comparison to d’Amico classification.
- “We chose not to include standard features such as PSA, clinical stage, GG on standard biopsy” à in order to offer a valid tool in management of prostate cancer, we should aim to the best AUC possible and not to “prove that imaging data combined with MRI-TB could be clinically relevant to predict post-RP outcomes”
- The short follow-up is an important limitation of the study as well reported by the authors.
Author Response
Dear Editor-in-Chief, Dear Reviewers,
Thank you so much for your work, your comments and your constructive criticisms.
We have duly considered your suggestions.
Our revisions are listed in this Letter:
Reviewer #1:
Several % in table 1 are not consistent (e.g. MRI stage and BCR)
REPLY We thank the reviewer for this comment. We corrected, sorry for the mistake.
Table 2 is hard to read in this fashion, I suggest to shift it in a vertical way in order to use a bigger font. REPLY We thank the reviewer for this comment. We shifted.
“We chose not to include standard features such as PSA, clinical stage, GG on standard biopsy” à in order to offer a valid tool in management of prostate cancer, we should aim to the best AUC possible and not to “prove that imaging data combined with MRI-TB could be clinically relevant to predict post-RP outcomes”
REPLY We thank the reviewer for this comment. We specified “The next step will be to repeat our investigation in a larger cohort of patients, with a longer follow-up and more events, the combination of all clinical, biological, radiological and pathological features into nomograms for BCR prediction purposes.” L 376-379
Reviewer #2:
When discussing prediction models of BCR, please refer to the recently proposed classification model of the EAU guidelines (EAU high vs low risk) published by Van den Broeck et al 2019 Eur urol. This is a validated model, based on clinical features and as this is recommended by the recent guidelines, it should be mentioned in the introduction.
REPLY We thank the reviewer for this comment. We added “Recent multivariate analysis [3] proposed new classification for patients experiencing BCR that differentiate patients with low or high risk of clinical progression based on PSA-doubling time, interval to biochemical failure and prostatectomy gleason score.” L 42-44.
Please provide a definition of 'clinical significant' prostate cancer
REPLY We thank the reviewer for this comment. We added ISUP > 1 or 2 depending on the study
Line 66: change " at least 2RB per suspicious lesion were taken"
REPLY We thank the reviewer for this comment. We changed L XX
You decided to report Grade Groups. Please use ISUP grade groups as this is the most recent recommendation to report histological staging.
REPLY We thank the reviewer for this comment. We changed.
In the statistical methods you mention performing uni- and multivariate analysis. For model design: you only used characteristics significant in MVA. However, these results are not clearly presented in the draft as it is now. I would recommend to show clear UVA and MVA tables.
Moreover, how were the factors included in the MVA selected? Based on significant results of the univariate analysis? Please specify.
REPLY We thank the reviewer for this comment. We only performed univariate analysis. Because of the low number of event, include the 4 factors significant in univariate analysis would not have been correct. We specified and deleted multivariate analysis in statistical method.
Why was diameter of the lesions not incorporated in the model?
REPLY We thank the reviewer for this comment. We have chosen not to incorporate the diameter of the lesions in our model due to the low availability of this information in current practice. In our cohort, this information was not available for almost 20% of the patients.
We added “”L 242-245
Of all patients with ECE preoperatively: only 80% had T3 or higher on pathological review. Does this mean overstaging by MRI in 20%? Please discuss this in the discussion session, since this is an interesting finding. Are you concerned that this overstaging negatively influences your model, since ECE is one of the three elements in your model.
REPLY We thank the reviewer for this comment. 11 patients suffered from overestimation by MRI. We added this limitation “Radiologists overstaged lesions for 11 patients, incorrectly claiming an extracapsular extension. Low-sensibility of extracapsular extension on MRI is known [31] but overestaging was not studied and could present as a limitation in our model.” L392-395
Validation cohort is not entirely independent since you used them to identify the factors of the final model. You should highlight this more clearly. Also I would suggest to add the KM analysis for both cohorts seperately (maybe as supplemental) as well as the ROC curves with AUC for the model in the two different cohort. Please specify the used statistical analysis in the methods section as well, as this is missing at the moment.
REPLY. Accordingly, we added supplemental figure and specified statistical analysis.
The final model used ECE, GG in TB and PIRADS. Two remarks:
how were the different cut-offs decided?
REPLY We thank the reviewer for this comment. Different cut-offs have been chosen based on their predictive value in univariable analysis, using the HR of each value. L241-245
for the high risk group: what about the PIRADS score--> is this any PIRADS or ? please specify
REPLY We thank the reviewer for this comment. We specified “regardless of the PIRADS”
This classification model is only useful if MRI and fusion biopsies are available. Not every centre has MRI available, or is used as much as we do. Classification based on clinical features will remain important there. This should be mentioned in the limitations.
REPLY We thank the reviewer for this comment. We mentioned “In addition, this model is only useful if MRI and fusion biopsy are available, wich according to the recommendations these are are part of the standard of care.” L383-384
It would be interesting to add your MRI based characteristics to a validated model, predictive of BCR, like d'Amico or EAU high/low risk for BCR and see if it can improve prediction of BCR. If MRI is not available, they can still use the 'limited' model with clinical features only.
REPLY We thank the reviewer for this comment. We specified “The next step will be to repeat our investigation in a larger cohort of patients, with a longer follow-up and more events, the combination of all clinical, biological, radiological and pathological features into nomograms for BCR prediction purposes.” L 376-379
@Several % in table 1 are not consistent (e.g. MRI stage and BCR)
REPLY We thank the reviewer for this comment. We corrected, sorry for the mistake.
Reviewer 2 Report
I would like to congratulate the authors on their work. It is indeed an interesting and actual topic, with still a lot of ongoing progress.
Please consider the following remarks:
INTRODUCTION:
- when discussing prediction models of BCR, please refer to the recently proposed classification model of the EAU guidelines (EAU high vs low risk) published by Van den Broeck et al 2019 Eur urol. This is a validated model, based on clinical features and as this is recommended by the recent guidelines, it should be mentioned in the introduction.
- Please provide a definition of 'clinical significant' prostate cancer
METHODS
- line 66: change " at least 2RB per suspicious lesion were taken"
- You decided to report Grade Groups. Please use ISUP grade groups as this is the most recent recommendation to report histological staging
RESULTS:
- in the statistical methods you mention performing uni- and multivariate analysis. For model design: you only used characteristics significant in MVA. However, these results are not clearly presented in the draft as it is now. I would recommend to show clear UVA and MVA tables.
- Moreover, how were the factors included in the MVA selected? Based on significant results of the univariate analysis? Please specify.
- Why was diameter of the lesions not incorporated in the model?
- Of all patients with ECE preoperatively: only 80% had T3 or higher on pathological review. Does this mean overstaging by MRI in 20%? Please discuss this in the discussion session, since this is an interesting finding. Are you concerned that this overstaging negatively influences your model, since ECE is one of the three elements in your model.
- Validation cohort is not entirely independent since you used them to identify the factors of the final model. You should highlight this more clearly. Also I would suggest to add the KM analysis for both cohorts seperately (maybe as supplemental) as well as the ROC curves with AUC for the model in the two different cohort. Please specify the used statistical analysis in the methods section as well, as this is missing at the moment.
- The final model used ECE, GG in TB and PIRADS. Two remarks:
- how were the different cut-offs decided?
- for the high risk group: what about the PIRADS score--> is this any PIRADS or ? please specify
DISCUSSION
- This classification model is only useful if MRI and fusion biopsies are available. Not every centre has MRI available, or is used as much as we do. Classification based on clinical features will remain important there. This should be mentioned in the limitations.
- It would be interesting to add your MRI based characteristics to a validated model, predictive of BCR, like d'Amico or EAU high/low risk for BCR and see if it can improve prediction of BCR. If MRI is not available, they can still use the 'limited' model with clinical features only.
Author Response

(The authors gave the same response as above.)

Round 2
Reviewer 1 Report
The manuscript has been revised and all previous comments have been taken in consideration and suggestions fulfilled.